# DATA EFFICIENT ANY TRANSFORMER-TO-MAMBA DISTILLATION VIA ATTENTION BRIDGE

## ABSTRACT

State-space models (SSMs) have emerged as efficient alternatives to Transformers for sequence modeling, offering superior scalability through recurrent structures. However, their training remains costly and the ecosystem around them is far less mature than that of Transformers. Moreover, the structural heterogeneity between SSMs and Transformers makes it challenging to efficiently distill knowledge from pretrained attention models. In this work, we propose **C**ross-architecture distillation via **A**ttention **B**ridge(**CAB**), a novel data-efficient distillation framework that efficiently transfers attention knowledge from Transformer teachers to state-space student models. Unlike conventional knowledge distillation that transfers knowledge only at the output level, CAB enables token-level supervision via a lightweight bridge and flexible layer-wise alignment, improving both efficiency and transferability. We further introduce flexible layer-wise alignment strategies to accommodate architectural discrepancies between teacher and student. Extensive experiments across vision and language domains demonstrate that our method consistently improves the performance of state-space models, even under limited training data, outperforming both standard and cross-architecture distillation methods. Our findings suggest that attention-based knowledge can be efficiently transferred to recurrent models, enabling rapid utilization of Transformer expertise for building a stronger SSM community.

## 1 INTRODUCTION

Linear RNNs (Mamba (Gu & Dao, 2024), RWKV (Peng et al., 2023)) have re-emerged as a promising alternative to attention-based models through recurrent state transitions. In contrast, Transformers use explicit attention to model long-range interactions with high expressiveness, but at the cost of quadratic computation and limited efficiency on long sequences (Katharopoulos et al., 2020). Among linear RNNs, Mamba exhibits strong long-range modeling with excellent runtime efficiency (Dao & Gu). However, its training remains computationally expensive, and its ecosystem is far less mature than that of Transformers. This highlights a challenge: *how can we cost-effectively transfer the rich inductive biases and knowledge of pretrained Transformers into Mamba architectures* (Fig. 1a)?

A straightforward approach is knowledge distillation (KD), where a pretrained Transformer guides the Mamba student. However, this naive strategy exhibits several limitations: (1) it lacks explicit transfer of attention-related inductive bias, weakening long-range dependency modeling (Li et al., 2024; Wang et al., 2020); (2) it results in weak gradient signals due to long backpropagation paths from the output layer, limiting learning efficiency in deep architectures (Romero et al., 2014; Sun et al., 2019); (3) it ignores architectural differences, resulting in suboptimal knowledge transfer (Lu et al., 2022; Abnar et al., 2020). Although SSMs lack explicit token interactions, their internal representations exhibit structural similarity to Transformer attention maps (Fig. 1b, left), motivating aligned transfer strategies. However, as shown in Fig. 1b, right, directly injecting Transformer attention into Mamba leads to performance collapse, underscoring the need for structure-aware knowledge transfer.

Recent work on bridging attention-based and SSM-based architectures still faces key limitations. MOHAWK (Bick et al., 2024) introduce a principled multi-stage alignment of full attention matrices and hidden states across architectures (Fig. 2a). Their high memory consumption and training cost make it impractical to scale this attention alignment to large training volumes(Bick et al., 2025).

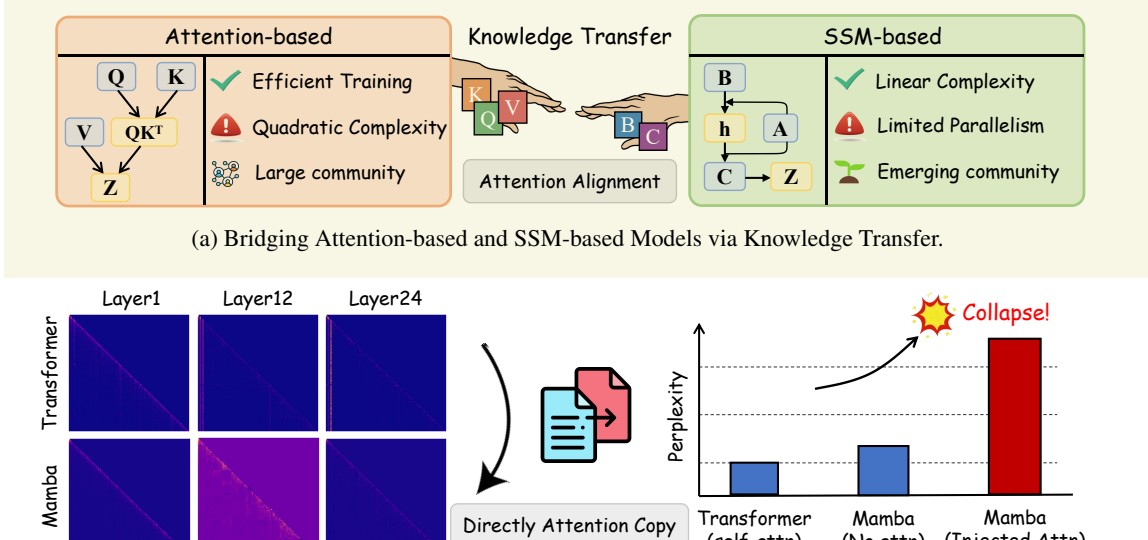

(a) Bridging Attention-based and SSM-based Models via Knowledge Transfer.

(b) Left: Attention maps in Transformer and Mamba show partial structural similarity. Right: Directly injecting Transformer attention into Mamba causes performance collapse.

Figure 1: **Towards Effective Attention-to-SSM Distillation.** We highlight the structural complementarity between attention-based and SSM-based models, and the limitations of direct attention transfer, motivating our proposed alignment-based distillation approach.

Wang et al. (Wang et al., 2024) attempts to connect Transformers and Mamba by reusing projection weights from Transformer layers(Fig. 2b), but direct substitution ignores input-dependent dynamics.

Furthermore, these prior works often assume access to full-scale training datasets, overlooking efficiency considerations. Attention models are notoriously data-hungry, requiring large amounts of data to learn robust relationships (Kaplan et al., 2020; Dosovitskiy et al.; Zhai et al., 2022), while SSMs struggle under limited supervision. Thus, guiding Mamba with pretrained Transformers becomes especially important for efficient learning in data-scarce real-world domains such as healthcare (Sheller et al., 2020; Li et al., 2019; Rieke et al., 2020), robotics (Kadian et al., 2020; Peng et al., 2018), and edge computing (Zhou et al., 2019; Satyanarayanan, 2017).

To address these limitations, we propose **CAB**, a novel framework for transferring attention-based inductive biases into Mamba models ( Fig. 2c). We introduce a lightweight MLP-based **bridge** between Transformer and Mamba architectures, enabling efficient fine-grained supervision of attention. Furthermore, we introduce a hierarchical mapping strategy to bridge the architectural gap between Transformers and Mamba, allowing for more effective cross-architecture knowledge transfer. Experiments on both vision and language modeling tasks demonstrate that our approach achieves superior performance and efficiency compared to existing distillation baselines.

Our contributions are summarized as follows:

- We introduce a novel distillation framework CAB, introduces a lightweight MLP **Attention Bridge** to directly connect the internal representations of the teacher and student models. This design enables fine-grained, token-level supervision and facilitates knowledge transfer across heterogeneous architectures.

- The proposed framework achieves dual efficiency: it is resource-efficient, bypassing the prohibitive cost of dense matrix alignment used in prior work, and uniquely data-efficient, designed to excel in low-data regimes where standard models falter.

- We extensively evaluate our method on both vision and language tasks between 9 diverse model architectures, consistently outperforming soft distillation and recent cross-architecture baselines.

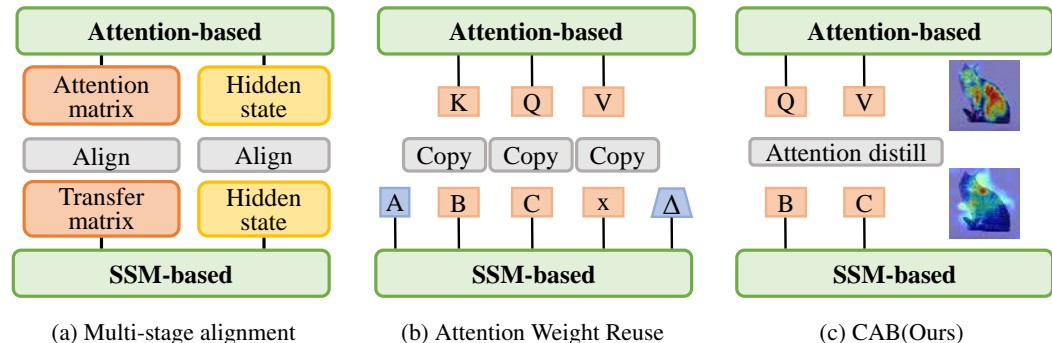

(a) Multi-stage alignment     (b) Attention Weight Reuse     (c) CAB(Ours)

Figure 2: Overview of representative strategies for transferring attention-based inductive biases from Transformers to State Space Models (SSMs).

## 2 RELATED WORK

**Linear Sequence Models.** Recent research has explored alternatives to traditional attention mechanisms for efficient sequence modeling. Linear attention mechanisms retain the query-key-value structure of Transformers but replace softmax with activation function mappings to achieve linear time complexity. Efficient Attention (Shen et al., 2021) was among the first to decouple the attention map computation into two linear projections, reducing time and memory costs. Performer (Choromanski et al.) approximates softmax attention using orthogonal random features with theoretical guarantees under linear complexity. CosFormer (Qin et al.) and Linear transformers (Katharopoulos et al., 2020) both adopt feature-mapped formulations of attention, introducing reweighting or recurrence structures that support causal and efficient computation.

In parallel, another line of work revisits linear recurrence as an alternative mechanism for sequence modeling, leading to renewed interest in linear RNNs and state space models. S4 (Gu et al.) introduced structured state matrices for efficient, parallel, and long-range sequence modeling. Mamba (Gu & Dao, 2024) extended this with input-dependent, token-wise state transitions. Mamba-2 (Dao & Gu) further improves speed and scalability through structured state-space duality (SSD) and hardware-efficient algorithm design. RWKVs (Peng et al., 2023; 2024; 2025) is a recent line of recurrent models that utilize Time-mix for temporal dependency modeling and Channel-mix for feature transformation.

**Knowledge Distillation and Attention Transfer.** Knowledge distillation (KD) aims to transfer learned knowledge from a large teacher model to a smaller student model. Originally proposed by Hinton et al. (Hinton et al., 2015) through logit-based distillation, KD has since been successfully applied across a variety of domains, including image classification (Romero et al., 2014)and language understanding (Sun et al., 2019). Within the Transformer framework, recent work has explored richer supervision signals beyond output logits. TinyBERT (Jiao et al., 2020) and MiniLM (Wang et al., 2020) distill attention distributions and intermediate hidden states, showing that aligning multi-level information improves downstream performance. Other works (Zagoruyko & Komodakis, 2017; Li et al., 2024) highlight the transferability of attention maps in both language and vision domains, suggesting that attention matrices encapsulate inductive biases useful for generalization.

However, most existing attention distillation methods assume that the teacher and student share similar Transformer architectures. In contrast, our work addresses the more challenging setting of cross-architecture distillation from Transformer teachers to state-space model students.

## 3 METHOD

### 3.1 PRELIMINARY

The standard attention mechanism models token interactions using a global attention matrix:

$$\text{Attention}(Q, K, V) = \text{softmax}\left(QK^\top\right) V, \tag{1}$$

where $Q, K, V$ represent the query, key, and value projections, respectively. While this formulation enables powerful context modeling, its quadratic time and space complexity with respect to sequence length $L$ significantly limits scalability in long-sequence tasks.

To address the inefficiency of standard attention, linear attention was proposed by approximating the softmax-based attention weights using activation function mappings:

$$\text{Attention}(Q, K, V) = \phi(Q)\varphi(K)^\top V, \tag{2}$$

where $\phi(\cdot)$ and $\varphi(\cdot)$ denote activation functions. Under causal constraints, this formulation admits a recursive structure:

$$h_t = h_{t-1} + \varphi(k_t)^\top v_t, \quad y_t = \phi(q_t)h_t, \tag{3}$$

where $h_t$ acts as a memory of past key–value information and $\phi(q_t)$ performs a dynamic readout. This reduces complexity to $\mathcal{O}(L)$ and makes linear attention structurally close to a recurrent model.

Mamba is a state-space model that encodes token interactions implicitly via dynamic recurrence:

$$y_t = C_t \sum_{j=1}^{t} \left( \Pi_{k=j+1}^{t} \bar{A}_k \right) \bar{B}_j x_j. \tag{4}$$

where the output $y_t$ aggregates all past inputs $\{x_j\}_{j=1}^{t}$ modulated by the transition matrices $(\bar{A}, \bar{B}_t, C_t)$. When the transition matrix $\bar{A}$ approximates the identity, this recurrence simplifies into a form structurally equivalent to the recursive formulation of linear attention:

$$\begin{cases} h_t = h_{t-1} + \varphi(k_t)^\top v_t, \\ y_t = \phi(q_t)h_t \end{cases} \iff \begin{cases} h_t = \bar{A}_t h_{t-1} + \bar{B}_t x_t, \\ y_t = C_t h_t \end{cases} \tag{5}$$

This perspective provides a principled bridge between attention and state-space models: under this approximation, aligning the token-dependent projections $B$ and $C$ with the Transformer's $K$ and $Q$ becomes a natural and theoretically motivated strategy. Prior work further shows that $Q, K$ encode sufficient inductive biases for transferring attention (Li et al., 2024; Clark et al., 2019). We therefore regard $B_t, C_t$ as **implicit attention carriers** and align them with the teacher's $Q, K$, providing fine-grained supervision without incurring quadratic overhead from dense attention maps.

## 3.2 ATTENTION BRIDGE

Inspired by recent representation alignment methods REPA (Yu et al.), we propose a lightweight *MLP-based bridge* that links the attention mechanisms of Transformer and Mamba. Specifically, the bridge maps $(Q, K)$ from the Transformer's explicit attention to $(B, C)$ in Mamba's implicit attention, adapting to input-dependent dynamics. This design enables *fine-grained, token-level supervision*: each token's role in the attention space is aligned with its counterpart in the state-space recurrence.

**Attention Alignment.** Existing attention distillation methods (Zagoruyko & Komodakis, 2017; Jiao et al., 2020) typically rely on explicit attention matrices as supervision targets. However, in state-space models like Mamba, attention is implicit and cannot be directly extracted without prohibitive quadratic overhead, as in full attention alignment methods (Bick et al., 2024). This gap motivates the need for a new distillation paradigm that leverages internal token-wise projections rather than explicit pairwise interactions. While $\bar{B} \leftrightarrow K$ and $C \leftrightarrow Q$ provide a bridge in the *discretized* SSM, we align the *continuous-time*, token-dependent $B$ and $C$ instead, to retain input-specific dynamics.

However, the representation spaces of $B, C \in \mathbb{R}^{L \times d_s}$ and the teacher's attention vectors $K, Q \in \mathbb{R}^{L \times d_t}$ are not inherently aligned—neither in **dimensionality** nor in **semantics**. To address this mismatch, we introduce learnable MLP-based projection modules that map the student's $B$ and $C$ into the teacher's attention space. These modules implicitly learn activation mappings and dimensional adaptation, enabling a flexible and effective form of attention-level knowledge transfer across heterogeneous architectures. The attention alignment loss is then defined as:

$$\mathcal{L}_{\text{attn}} = \frac{1}{L} \sum_{l=1}^{L} \left( \left\| \phi_B(B^{(l)}) - K^{(l)} \right\|_2^2 + \left\| \phi_C(C^{(l)}) - Q^{(l)} \right\|_2^2 \right), \tag{6}$$

where $l$ indexes student layers and $\phi_B, \phi_C$ are MLPs. This alignment avoids explicit attention maps and supports distillation under limited supervision, making it suitable for low-data scenarios.

**Layer-wise Alignment across Architectures.** A key challenge in cross-architecture distillation lies in the structural mismatch between attention-based models and SSMs. Attention-based teachers often contain deep stacks of self-attention layers, while student models based on state-space architectures may have a different number of layers due to design choices. This mismatch in depth makes strict one-to-one layer supervision infeasible and potentially suboptimal.

To address this, we adopt a flexible alignment strategy that allows cross-layer matching. Instead of enforcing strict one-to-one correspondence, we define a general layer mapping function $g(l)$ that aligns each student layer $l \in [1, L]$ to a teacher layer via proportional indexing:

$$g(l) = \left\lfloor \frac{l}{L} \cdot T \right\rfloor. \tag{7}$$

This unified and relaxed alignment strategy supports both deeper and shallower student networks, enabling effective knowledge transfer across varying depths and architectural module types while avoiding over-constraining the student. Moreover, it enhances the transferability of the method across heterogeneous architectures. The resulting attention alignment loss is:

$$\mathcal{L}_{\text{attn}} = \frac{1}{L} \sum_{l=1}^{L} \left( \left\| \phi_B(B^{(l)}) - K^{(g(l))} \right\|_2^2 + \left\| \phi_C(C^{(l)}) - Q^{(g(l))} \right\|_2^2 \right). \tag{8}$$

## 4 EXPERIMENT

### 4.1 SETUP

We evaluate our proposed attention distillation framework across both vision and language domains, focusing on simulating low-resource scenarios to study data efficiency and cross-architecture transferability. For image classification, we use the ImageNet-1k dataset (Deng et al., 2009), and simulate realistic low-resource settings by training on 1%, 5%, 10%, or 20% of the original training data, sampled per class to preserve label balance. All models are evaluated on the full validation set. For language modeling, we use the OpenWebText corpus (Gokaslan et al., 2019) with a maximum sequence length of 1024 tokens for distillation. To simulate supervision-limited regimes, we adopt a two-stage training strategy: the first stage uses 200M tokens for attention alignment, and the second stage scales up to 2B or 4B tokens for soft distillation. We evaluate models directly using perplexity on OpenWebText, C4 (Raffel et al., 2020), and WikiText (Merity et al., 2016) datasets, to reflect generalization under pretraining-only supervision.

**Implementation Details.** We implement each projection module $\phi_B$ and $\phi_C$ as a 2-layer MLP with SiLU activation functions across all experiments, which we find to be both simple and effective. For image classification, we use DeiT (Touvron et al., 2021) as the Transformer teacher and Vision Mamba (Zhu et al.) as the state-space student model. As shown in equation 5, initializing $A \approx 0$ (so that $\bar{A} \approx I$) simplifies the recurrence into a form structurally equivalent to linear attention, making it natural to align $B$ and $C$ with the teacher's $K$ and $Q$ through our attention bridge.

Given the bidirectional nature of Vision Mamba, we compute the attention-level distillation loss separately for the forward and backward directions. Specifically, for each direction dir $\in$ {forward, backward}, the loss is computed as:

$$\mathcal{L}_{\text{dir}} = \frac{1}{L} \sum_{l=1}^{L} \left( \left\| \phi_B(B_{\text{dir}}^{(l)}) - K^{(g(l))} \right\|_2^2 + \left\| \phi_C(C_{\text{dir}}^{(l)}) - Q^{(g(l))} \right\|_2^2 \right), \quad \text{dir} \in \{\text{forward, backward}\}. \tag{9}$$

The total distillation loss is the sum of both directions:

$$\mathcal{L}_{\text{attn}} = \mathcal{L}_{\text{forward}} + \mathcal{L}_{\text{backward}}. \tag{10}$$

The teacher models are DeiT-Tiny and DeiT-Small, both based on Transformer architectures, while the student models are Vim-Tiny and Vim-Small, based on state-space mechanisms. Table 1 summarizes the architectural specifications of the teacher and student models, including their modeling mechanisms, number of layers, and parameter counts.

Table 1: Architectural specifications of vision models.

| Model | Hidden Dim | Layers | Params (M) |
|---|---|---|---|
| DeiT-Tiny | 192 | 12 | 5 |
| DeiT-Small | 384 | 12 | 22 |
| Vim-Mini* | 96 | 12 | 1.1 |
| Vim-Mini | 96 | 24 | 2.1 |
| Vim-Tiny* | 192 | 12 | 3.8 |
| Vim-Tiny | 192 | 24 | 7.1 |
| Vim-Small | 384 | 24 | 26 |

Table 2: Architectural specifications of language models.

| Model | Layers | Params | FLOPs (G) |
|---|---|---|---|
| DistilGPT2 | 6 | 88M | 20.76 |
| Phi-Mamba | 6 | 123M | 21.68 |

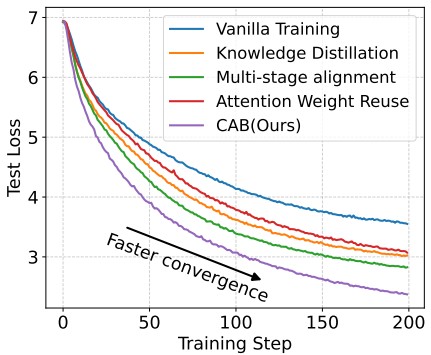

Figure 3: Convergence speed and test loss comparison on ImageNet.

All ViM models are distilled for 300 epochs using the AdamW optimizer with a learning rate of 5e-4. The batch size is 64, and training is conducted on a single NVIDIA A100 GPU. Standard ImageNet augmentations, including random cropping and horizontal flipping, are applied during training.

For all language experiments, we use a lightweight Phi-Mamba-123M variant (Bick et al., 2024) as the student model and DistilGPT2 (pretrained on OpenWebText) as the Transformer teacher. Built on Mamba-2, Phi-Mamba parameterizes $A_i$ as a scalar $a_i$, making it naturally close to linear attention; thus we directly align $B$ and $C$ without special initialization. To isolate the contribution of our attention alignment mechanism, training is conducted in two stages: (i) attention alignment, where the student's token-wise projections are matched to the teacher's key and query representations (Eq. 8); and (ii) soft distillation, minimizing KL divergence between teacher and student outputs. Both stages share the same configuration: learning rate $2 \times 10^{-5}$, batch size 32 per device, and 8 NVIDIA A100 GPUs. Evaluation is performed in terms of perplexity (PPL) on OpenWebText, C4 (Raffel et al., 2020), and WikiText (Merity et al., 2016). Table 2 summarizes the architectural specifications of these language models, including their parameter counts and FLOPs.

**Baselines.** We compare our method against several related distillation strategies:

- **Standard Soft Distillation** (Hinton et al., 2015): The student model is trained to match the teacher's softened output logits using KL divergence loss, without supervision on intermediate representations.

- **Mamba in Llama(Attention Weight Reuse)** (Wang et al., 2024): This method distills large language models (LLMs) into hybrid Transformer-Mamba architectures by reusing attention weights. We adapt their strategy to our vision and language settings.

- **MOHAWK(Multi-Stage Alignment)** (Bick et al., 2024): This approach distills Transformers into state-space models by progressively aligning full attention matrices as well as intermediate hidden representations. We reimplement their method and apply it in our cross-architecture setting.

Implementation and architectural details of all baselines, including training configurations and model specifications, are provided in Appendix C.

## 4.2 MAIN RESULTS

**Image Classification.** Table 3 summarizes Top-1 accuracy on ImageNet for various teacher-student pairs and data ratios. Our method consistently outperforms the standard soft distillation and other baselines across these settings. Notably, under the DeiT-Tiny teacher and Vim-Tiny student setting at 10% data, it achieves up to a **+16.3%** improvement over vanilla training. Beyond final accuracy, we also analyze the convergence dynamics of different training strategies. Fig. 3 presents these results. Our method converges significantly faster and achieves lower test loss throughout training, highlighting its optimization efficiency under limited supervision. For completeness, we also report

Table 3: Top-1 accuracy comparison between pretraining and distillation methods on ImageNet classification under varying proportions of training data.

| Teacher | Student | Method | 1% | 5% | 10% | 20% |
|---|---|---|---|---|---|---|
| - | Vim-Tiny | Vanilla Training | 11.2 | 27.4 | 32.9 | 41.5 |
| DeiT-Tiny | Vim-Tiny | Soft Distillation | 20.4 | 37.0 | 42.0 | 42.7 |
| | | Attention Weight Reuse | 21.3 | 37.1 | 41.5 | 42.6 |
| | | Multi-Stage Alignment | 21.5 | 36.8 | 45.1 | 44.0 |
| | | **CAB**(Ours) | **27.8** | **45.4** | **49.2** | **49.4** |
| - | Vim-Small | Vanilla Training | 15.3 | 29.2 | 36.2 | 47.0 |
| DeiT-Small | Vim-Small | Soft Distillation | 23.3 | 41.1 | 49.4 | 54.0 |
| | | Attention Weight Reuse | 23.5 | 41.5 | 49.3 | 54.3 |
| | | Multi-Stage Alignment | 24.0 | 42.1 | 50.1 | 53.9 |
| | | **CAB**(Ours) | **27.3** | **46.0** | **54.9** | **60.7** |

results under the full-data regime, where a DeiT-Tiny teacher distills into a Vim-Tiny student using 100% of ImageNet; see Appendix A.1.

**Attention Behavior Analysis.** To further understand the underlying attention behaviors, we analyze the similarity between attention matrices of the Vim model and pretrained ViT across all layers(Fig. 4). Before L3, similarity slightly decreases, reflecting the local focus of early layers (Zeiler & Fergus, 2014). Beyond L3, however, similarity rises sharply, showing that our alignment loss effectively guides Vim to mimic Transformer-style attention. This indicates that mid-to-deep layers recover global token interactions typically lost in standard training, thereby bridging the gap between SSMs and attention-based models.

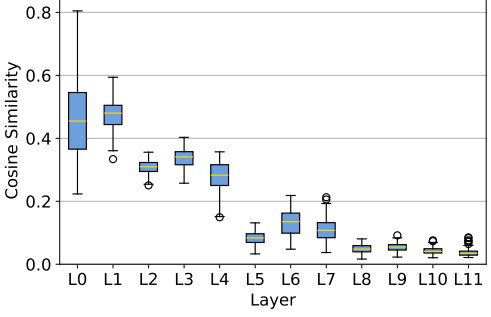

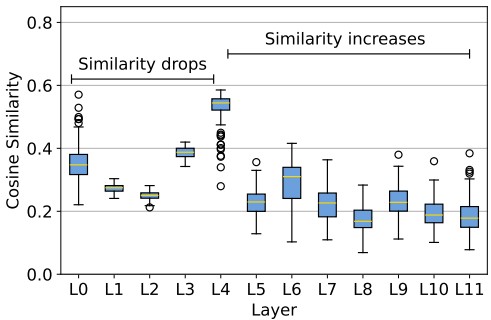

(a) Similarity between pretrained ViT and Vim with vanilla training

(b) Similarity between pretrained ViT and Vim with attention alignment.

Figure 4: Cosine similarity between Vim and pretrained ViT, comparing results with and without attention alignment. Higher similarity indicates better alignment of attention representations.

**Language Modeling.** Table 4 reports perplexity results evaluated under two training regimes corresponding to the second training stage, using 2B and 4B tokens respectively. Our proposed attention-level distillation consistently achieves lower perplexity across all benchmarks and scales, significantly outperforming both standard attention weight reuse and multi-stage alignment baselines. Remarkably, with only 4B tokens, the CAB-distilled student attains perplexity comparable to the Transformer teacher, highlighting the efficiency of our bridge in transferring knowledge from attention-based to state-space models.

Beyond achieving strong performance on the in-distribution OpenWebText corpus used for distillation, our method generalizes effectively to diverse out-of-distribution (OOD) datasets such as C4 and WikiText, which differ markedly in domain and style. Notably, the relative perplexity reduction on

Table 4: Perplexity comparison on language modeling benchmarks. DistilGPT2 is used as the teacher and Phi-Mamba as the student.

| Method | OpenWebText | | C4 | | WikiText | |
|---|---|---|---|---|---|---|
| | 2B | 4B | 2B | 4B | 2B | 4B |
| *Teacher PPL (DistilGPT2)* | *28.2* | | *39.7* | | *52.96* | |
| Attention Weight Reuse | 61.8 | 37.2 | 111.0 | 62.9 | 212.3 | 99.8 |
| Multi-Stage Alignment | 58.4 | 31.1 | 105.7 | 51.2 | 199.3 | 77.9 |
| **CAB (Ours)** | **54.4** | **30.1** | **97.3** | **50.1** | **175.0** | **74.7** |

these challenging OOD datasets is larger than that on OpenWebText, indicating that our framework successfully transfers underlying structural knowledge beyond the training distribution. This enables robust language modeling without task-specific fine-tuning and highlights the broad practical applicability of our approach in real-world scenarios. For completeness, Appendix A.2 reports scalability results with Phi-1.5 as the teacher and Phi-Mamba-1.5B as the student.

**Efficiency Analysis.** In addition to perplexity and generalization results, we further evaluate the *efficiency* of CAB compared to MOHAWK. At sequence length $L = 1024$, aligning a single attention layer with MOHAWK requires storing $2 \cdot (H, L, L)$ tensors, leading to quadratic memory growth and substantial runtime overhead. By contrast, CAB avoids explicit attention matrices and only introduces two lightweight MLPs of size $2 \cdot n_{\text{MLP layers}} \cdot (d_{\text{student}}, d_{\text{teacher}})$, whose cost scales linearly with model dimensions rather than sequence length. On DistilGPT2 $\rightarrow$ Phi-Mamba-123M with 200M tokens, CAB achieves a $10\times$ reduction in memory footprint and $4\times$ faster runtime on a single A100 GPU, as shown in Fig. 5. These results highlight that CAB not only improves accuracy but also makes large-scale distillation much lighter and easier to run in practice.

### 4.3 ABLATION STUDY

To evaluate the effectiveness and robustness of our proposed distillation framework, we conduct ablation studies from three complementary perspectives: (1) the contribution of $B$ and $C$ projection alignment to knowledge transfer, (2) the impact of the transition matrix $\bar{A}$ initialization strategy, and (3) the generalization of our method across student architectures with varying capacities. All ablation experiments are conducted under the DeiT-Tiny teacher and Vim-Tiny student setting on a fixed 10% subset of ImageNet-1k to reduce computational cost while maintaining task difficulty.

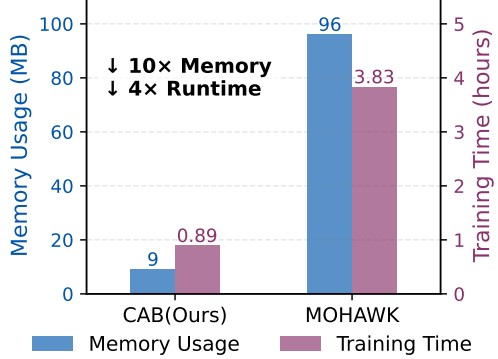

Figure 5: Memory and runtime comparison of CAB vs. full-matrix distillation.

**Effect of $B/C$ Distillation.** We first investigate whether aligning both $B$ and $C$ projections is necessary, or if aligning either alone suffices. We compare three variants: distilling only $B$, only $C$, and both. As shown in Table 5, we observe that while distilling either component individually brings measurable gains over no distillation, jointly aligning $B$ and $C$ yields the best performance. This supports our hypothesis that $B$ and $C$ jointly encode the implicit attention mechanism in Mamba, and both are needed to effectively transfer structural knowledge from the teacher. We also test a variant where $B$ and $C$ share the same projection ($\phi_B \equiv \phi_C$). As shown in Table 5, this setting underperforms independent mappings, since $B$ and $C$ lie in different semantic spaces—$B$ injects inputs while $C$ extracts context—confirming the need for distinct projections.

**Effect of $\bar{A}$ Initialization.** We study how the initialization of the transition matrix influences the effectiveness of distillation. In Mamba, $A$ is typically initialized with a log-spaced diagonal

from S4D (Gu et al., 2022), which is designed to capture diverse temporal dynamics in sequence modeling. However, such initialization introduces dynamics that may not align well with attention-based supervision. In our setting, we instead initialize $A \approx 0$, so that $\bar{A} \approx I$, effectively simplifying the recurrence into an additive form that closely resembles linear attention. This choice provides a better structural match to Transformer teachers, reducing the gap between the two architectures. As a result, the attention-aligned initialization not only stabilizes optimization during training but also improves transfer efficiency. Empirically, as shown in Table 5, it consistently outperforms the default S4D initialization across tasks, highlighting the importance of initialization when bridging heterogeneous sequence models.

**Effect of Student Architecture.** We evaluate the robustness of our framework across student models with varying depths and widths. This setup reflects realistic cross-architecture scenarios where students may differ significantly from teachers in capacity and design. As shown in Table 6, our method consistently improves performance across diverse architectures. Even under aggressive compression (e.g., shallow or narrow students), attention-level distillation yields clear gains over standard knowledge distillation, demonstrating the versatility of our framework for heterogeneous student designs.

Table 5: Ablation study on distillation strategys.

| Setting | Acc (%) |
|---|---|
| Vanilla Training | 32.9 |
| Soft Distillation | 42.0 |
| Align $B$ only | 48.7 |
| Align $C$ only | 48.8 |
| Shared $\phi$ for $B$ and $C$ | 49.0 |
| Align $B + C$ w/ default $\bar{A}$ | 43.1 |
| Align $B + C$ w/ $\bar{A} \approx I$ (Ours) | **49.2** |

Table 6: Top-1 accuracy comparison between soft distillation and our method across various student architectures and proportions of training data.

| Teacher | Student | Params (M) | Method | 1% | 5% | 10% | 20% |
|---|---|---|---|---|---|---|---|
| DeiT-Tiny | Vim-Mini* | 1.1 | Soft Distillation | 5.1 | 17.6 | 30.3 | 30.0 |
| | | | **CAB**(Ours) | **6.9** | **18.7** | **31.0** | **32.8** |
| | Vim-Mini | 2.1 | Soft Distillation | 6.6 | 24.0 | 34.6 | 35.3 |
| | | | **CAB**(Ours) | **8.0** | **24.1** | **36.0** | **36.2** |
| DeiT-Small | Vim-Tiny* | 3.8 | Soft Distillation | 7.7 | 27.9 | 42.3 | 41.5 |
| | | | **CAB**(Ours) | **8.1** | **28.1** | **42.8** | **42.5** |
| | Vim-Tiny | 7.1 | Soft Distillation | 7.8 | 27.6 | 43.9 | 45.0 |
| | | | **CAB**(Ours) | **8.7** | **30.7** | **44.9** | **47.2** |

## 5 CONCLUSION

In this work, we propose **CAB**, a novel attention distillation framework for transferring structural knowledge from Transformer-based teachers to state-space student models. CAB introduces a lightweight MLP-based **Attention Bridge** that treats the token-dependent projections in Mamba as implicit carriers of attention and aligns them with the teacher's key and query representations. Extensive experiments across vision and language tasks demonstrate that this strategy significantly improves the performance of state-space models.

We believe our approach offers a new perspective on cross-architecture knowledge transfer and highlights the potential of combining attention-based reasoning with efficient recurrent modeling. Importantly, we demonstrate that this framework remains highly effective under data-scarce conditions. Through extensive experiments on subsets of ImageNet and language modeling tasks, we show that our method achieves strong performance with only 1%–20% of the training data, significantly outperforming existing distillation baselines. Our findings highlight a promising direction toward building efficient, scalable, and data-efficient sequence models. However, our approach currently focuses on Mamba and Transformers, and extending it to other SSM variants or more complex hybrid architectures may require further investigation.

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

## LLM USAGE

Large language models (LLMs) were used solely to polish the writing (e.g., grammar correction and phrasing improvements). They did not contribute to research ideation.

## A    ADDITIONAL EXPERIMENTS

### A.1    FULL-DATA REGIME ON IMAGENET

We further investigate CAB under the full-data regime, where a DeiT-Tiny teacher distills into a **Vim-Tiny** student using 100% of ImageNet. As shown in Fig. 6 and Table 7, CAB dramatically accelerates convergence in the early stages, even surpassing standard soft distillation. However, when enforced throughout training (e.g., 300 epochs), CAB alignment degrades performance, suggesting that excessive attention supervision can over-constrain the mismatch between Transformer attention and the SSM latent space. Inspired by recent findings on (Wang et al., 2025), we adopt an *early-stopped alignment* strategy: applying CAB for the first phase and then switching to KL distillation. Here, **(xepoch) denotes the epoch at which CAB alignment is early-stopped**. This hybrid schedule successfully transfers attention priors while maintaining strong final accuracy.

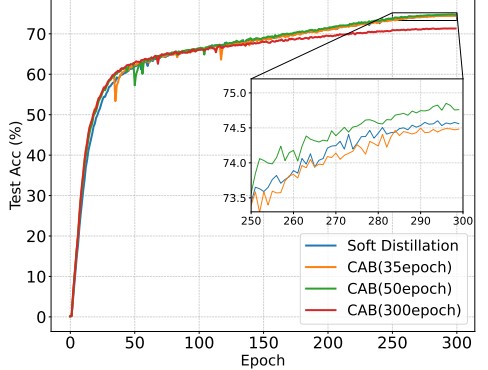

Figure 6: Training curves under full-data regime.

| Method | Top-1 Acc ↑ | Top-5 Acc ↑ |
|---|---|---|
| DeiT-Tiny(teacher) | 72.20 | 91.10 |
| Soft Distillation | 74.56 | 92.46 |
| CAB(35epoch) | 74.50 | 92.38 |
| CAB(50epoch) | **74.85** | **92.56** |
| CAB(300epoch) | 71.34 | 90.64 |

Table 7: Results under the full-data regime.

### A.2    SCALABILITY TO LARGER LLMS

To further assess scalability, we conduct experiments using **Phi-1.5B** as the Transformer teacher and **Phi-Mamba-1.5B** as the student. We re-implemented the training pipeline of (Bick et al., 2024) according to the descriptions provided in the paper. However, since the original setting does not specify dataset splits and cannot be exactly reproduced, we empirically determine the splits and adopt a two-stage training strategy for a fair comparison and isolate the contribution of our attention alignment mechanism: the first stage uses 200M tokens for attention alignment, and the remaining 2.8B tokens are allocated to the KD stage. All models are trained on the C4 corpus (Raffel et al., 2020) with a maximum sequence length of 2048 tokens. Both stages use a learning rate of 5e-5, a batch size of 32 per device, and are trained on 8 NVIDIA A100 GPUs. This setting is evaluated on a suite of downstream language modeling benchmarks using the LM Harness. Results demonstrate that CAB consistently improves perplexity across diverse benchmarks, validating its effectiveness beyond lightweight configurations. Table 8 summarizes the results. Under the same training setup, our proposed **CAB** consistently outperforms prior distillation methods.

## B    DISCUSSION WITH MORE RELATED WORKS

A growing body of work has explored knowledge distillation across different architectural families, aiming to leverage the strengths of large pretrained models while deploying lightweight or domain-specialized students. In the NLP domain, TinyBERT (Jiao et al., 2020), MiniLM (Wang et al., 2020), and MobileBERT (Sun et al., 2020) focus on intra-Transformer distillation, transferring attention and

Table 8: Benchmark comparison on five tasks

| Model | HellaSwag | PIQA | ARC-E | ARC-C | WinoGrande |
|---|---|---|---|---|---|
| Phi1.5 | 62.6 | 75.6 | 73.1 | 48.0 | 72.9 |
| Attention Weight Reuse | 29.42 | 62.79 | 36.07 | 21.84 | 51.22 |
| MOHAWK | 28.87 | 63.11 | 36.24 | 22.70 | 51.22 |
| **CAB (Ours)** | **31.73** | **64.64** | **37.75** | **23.21** | **51.85** |

hidden representations within the same architectural paradigm. Beyond homogeneous architectures, **cross-structure distillation** has attracted growing interest. Early works such as Patient Knowledge Distillation (Sun et al., 2019) and FitNets (Romero et al., 2014) employed intermediate supervision to facilitate knowledge transfer across networks of varying depth or capacity. ConvBERT (Jiang et al., 2020) introduced span-based dynamic convolution into Transformers, improving local feature modeling while remaining compatible with attention-based distillation. Liu et al. (Liu et al., 2022) proposed a unified cross-architecture distillation framework by learning token-level correspondences and relational structure mappings from Transformers to CNNs. However, most of these approaches rely on feature-level alignment or final-layer supervision, without explicitly preserving attention-level semantics when transferring across fundamentally different inductive biases.

While existing approaches have made progress in distilling from Transformers to Mamba-like models, several challenges remain unresolved. A key limitation is the lack of a unified mechanism for transferring attention-based relational structure to models with fundamentally different inductive biases. TransMamba (Chen et al., 2025) sidesteps the attention mechanism entirely, relying on hierarchical feature calibration, while Bick et al. (Bick et al., 2024) impose handcrafted architectural constraints to align internal representations. However, such design choices are model-specific and difficult to generalize across heterogeneous architectures. Wang et al. (Wang et al., 2024) introduces projection reuse by directly copying Transformer weights, but this static substitution ignores input-dependent dynamics inherent to SSMs. These methods, while valuable, fall short of addressing the core challenge of distillation across structurally dissimilar models without relying on manual architecture tailoring.

In contrast, our method directly bridges attention and state projections by introducing a representation-level alignment between $QK$ and $BC$ within each layer. This avoids architectural entanglement and allows for a clean formulation of attention-level distillation losses, which can be flexibly applied to discrete SSMs without modifying their internal propagation rules. Moreover, by initializing the transition matrix $A$ to near-zero, we induce a recurrence that behaves analogously to linear attention, enabling a natural path for knowledge flow from Transformer activations into Mamba dynamics. Our approach thus provides an interpretable, computation-friendly, and model-agnostic framework for attention-to-SSM distillation. It highlights a missing middle ground between full structural replication and output supervision, offering a scalable route to adapt pretrained Transformer knowledge into sequence models with fundamentally different inductive priors.

## C  EXPERIMENT SETTINGS

In this section, we detail the experimental settings and provide instructions for reproduction.

### C.1  IMPLEMENTATION DETAILS FOR VISION DISTILLATION METHODS

To ensure a fair and comprehensive comparison, we extend two representative Transformer-to-SSM distillation frameworks (Bick et al., 2024; Wang et al., 2024)to the vision domain. While both were originally proposed for language modeling, we adapt them for image classification on ImageNet-1k by modifying the distillation targets and model-specific components as follows:

**Transformers to SSMs (MOHAWK).**  The original MOHAWK framework (Bick et al., 2024) consists of three stages: attention matrix alignment, hidden state alignment, and output-level knowledge distillation. For fair comparison and reduced overhead, we adopt only the attention alignment and

final distillation stages. In the original setup, attention alignment was limited to 200M tokens (out of 3B total) due to the high cost of computing softmax attention.

In our adaptation to the vision setting, we perform one full epoch of distillation using the training set. The attention matrices from the Vim student are computed following the formulation used in *The Hidden Attention of Mamba Models* (Ali et al., 2024). In the second stage, we apply KL divergence between the teacher and student logits as the distillation objective. To handle depth mismatches between the teacher and student, we introduce a layer alignment function $g(l)$ that maps each student layer $l$ to its corresponding teacher layer. Hidden state matching is omitted to isolate the effect of attention transfer and maintain efficiency.

**Mamba in the Llama.**    Mamba in the Llama (Wang et al., 2024) introduces an attention-initialized Mamba variant, using a linearized formulation as the initialization scheme for distillation. Specifically, the linear projections of the Transformer teacher—including $Q$, $K$, $V$, and the output projection—are mapped to the corresponding $C$, $B$, $X$, and output projection layers in the Mamba student.

In our vision adaptation, we adopt the same strategy: the $Q$, $K$, $V$, and output projection matrices from the ViT teacher are linearly mapped to the $C$, $B$, $X$, and output projection modules in the Vim student. To address the mismatch in depth between teacher and student models, we further introduce our layer alignment function $g(l)$ to assign supervision across corresponding layers.

**CAB(Ours).**    We implement our bidirectional distillation framework tailored for visual tasks. Specifically, we align both the forward and backward dynamic projections $B$, $C$ from the student Vim with the $K$, $Q$ representations from the Transformer teacher (e.g., DeiT). This alignment is performed layer-wise via a mapping function $g(l)$ and optimized jointly with soft KL loss on logits. To improve compatibility between attention and state-space recurrence, we also initialize the Mamba transition matrix $A \approx 0$ to simulate linear attention behavior. A full description of the algorithm is provided in Algorithm 1.

---

**Algorithm 1:** Attention Distillation for Bidirectional Vision Mamba

---

**Input** : Transformer teacher $\mathcal{T}$ (e.g., DeiT), bidirectional Mamba student $\mathcal{S}$, image tokens $\{x_t\}_{t=1}^{L}$, loss weights $\lambda$
**Output** : Trained student model $\mathcal{S}$
1  Initialize Mamba transition matrix $A \approx 0$ such that $\bar{A} \approx I$ ;
2  Define student-to-teacher layer mapping $g(l)$ ;
3  **for** *each batch $\{x_t\} \sim \mathcal{D}$* **do**
4  $\quad \{K^{(l)}, Q^{(l)}\} \leftarrow \mathcal{T}(\{x_t\})$ ;            // extract key/query from teacher
5  $\quad \{B_{\text{fw}}^{(l)}, C_{\text{fw}}^{(l)}, B_{\text{bw}}^{(l)}, C_{\text{bw}}^{(l)}\} \leftarrow \mathcal{S}(\{x_t\})$ ;  // extract projections from student
6  $\quad$ **for** *each layer $l = 1$ to $L$* **do**
7  $\quad\quad \mathcal{L}_{\text{fw}}^{(l)} \leftarrow \|\phi_B(B_{\text{fw}}^{(l)}) - K^{(g(l))}\|^2 + \|\phi_C(C_{\text{fw}}^{(l)}) - Q^{(g(l))}\|^2$ ;
8  $\quad\quad \mathcal{L}_{\text{bw}}^{(l)} \leftarrow \|\phi_B(B_{\text{bw}}^{(l)}) - K^{(g(l))}\|^2 + \|\phi_C(C_{\text{bw}}^{(l)}) - Q^{(g(l))}\|^2$ ;
9  $\quad \mathcal{L}_{\text{attn}} \leftarrow \frac{1}{L}\sum_{l=1}^{L}(\mathcal{L}_{\text{fw}}^{(l)} + \mathcal{L}_{\text{bw}}^{(l)})$, $p_T \leftarrow \mathcal{T}(\{x_t\})$, $p_S \leftarrow \mathcal{S}(\{x_t\})$
10  $\quad \mathcal{L}_{\text{KL}} \leftarrow \text{KL}(p_T \| p_S)$ ;
11  $\quad$ Update $\mathcal{S}$ using $\nabla(\mathcal{L}_{\text{attn}} + \lambda \mathcal{L}_{\text{KL}})$ ;

---

Table 11 summarizes the training configurations for visual distillation experiments under different data regimes (1%, 5%, 10%, 20%). Models are trained on ImageNet-1k using BF16 precision on a single NVIDIA A100 GPU.

### C.2    IMPLEMENTATION DETAILS FOR LLM DISTILLATION METHODS

For the language modeling setting, we adapt both the **Transformers to SSMs (MOHAWK)** and **Mamba in the Llama** baselines into our distillation pipeline using the Phi-Mamba architecture as the student.

**Transformers to SSMs (MOHAWK).** For MOHAWK, we follow its original two-stage design by performing attention matrix alignment followed by output-level knowledge distillation. Specifically, we distill attention representations using 200M tokens in the first stage, and continue soft KL divergence-based distillation over 2B or 4B tokens depending on the experimental setup. To ensure fairness and training feasibility, we omit the intermediate hidden state alignment stage. Layer mismatches between the Transformer teacher (DistilGPT2) and the student are handled using a layer alignment function $g(l)$.

**Mamba in the Llama.** For Mamba in the Llama, we replicate their projection initialization strategy by directly mapping the Transformer's $Q$, $K$, $V$, and output projection weights to the Mamba's $C$, $B$, input, and output projection layers, respectively. We then train the student using the same two-stage strategy on 2B or 4B tokens without additional supervision.

**CAB(Ours).** We apply our attention-level distillation framework to the LLM setting. The student model (Phi-Mamba) is trained in two stages: first aligning the token-wise $B$, $C$ projections to the teacher's $K$, $Q$ via layer-mapped MLPs, followed by KL-based soft distillation on teacher logits. Detailed pseudocode for this training process is provided in Algorithm 2.

---

**Algorithm 2:** Two-Stage Attention Distillation for Mamba in Language Modeling

---

**Input** : Pretrained Transformer teacher $\mathcal{T}$, Causal Mamba student $\mathcal{S}$, tokenized dataset $\mathcal{D}$, total training tokens $N$, attention distillation loss $\mathcal{L}_{\text{attn}}$, soft distillation loss $\mathcal{L}_{\text{KL}}$

**Output** : Trained student model $\mathcal{S}$

1 **Stage 1: Attention Alignment Phase (first $N_1$ tokens)**

2 **for** *each batch* $(x, y) \sim \mathcal{D}$ **do**

3      $\{K^{(l)}, Q^{(l)}\} \leftarrow \mathcal{T}(x)$ ;          `// extract key/query from teacher`

4      $\{B^{(l)}, C^{(l)}\} \leftarrow \mathcal{S}(x)$ ;          `// extract projections from student`

5      **for** *each layer* $l = 1$ *to* $L$ **do**

6          Compute loss: $\mathcal{L}_{\text{attn}}^{(l)} = \|\phi_B(B^{(l)}) - K^{(g(l))}\|^2 + \|\phi_C(C^{(l)}) - Q^{(g(l))}\|^2$

7      $\mathcal{L} \leftarrow \frac{1}{L} \sum_{l=1}^{L} \mathcal{L}_{\text{attn}}^{(l)}$ Update student $\mathcal{S}$ using $\nabla \mathcal{L}$

8 **Stage 2: Soft Distillation Phase (continue to $N_2$ tokens)**

9 **for** *each batch* $(x, y) \sim \mathcal{D}$ **do**

10      $p_T \leftarrow \mathcal{T}(x)$ ;          `// teacher soft predictions`

11      $p_S \leftarrow \mathcal{S}(x)$ ;          `// student predictions`

12      Compute loss: $\mathcal{L}_{\text{KL}} = \text{KL}(p_T \| p_S)$

13      Update student $\mathcal{S}$ using $\nabla \mathcal{L}_{\text{KL}}$

---

Table 9 summarizes the training configurations for language model distillation. All experiments are conducted using BF16 precision on a single NVIDIA A100 GPU.

Table 9: Training recipe for LLM distillation experiments.

| Config | Value |
|---|---|
| Max sequence length | 1024 |
| Batch size | 4 |
| Learning rate | 2e-5 |
| Optimizer | AdamW |
| Adam $\beta$ | (0.9, 0.999) |
| Adam $\epsilon$ | 1e-8 |
| Precision | bfloat16 (BF16) |
| Training tokens | 200M (Stage 1) / 2B or 4B (Stage 2) |
| Loss functions | Attention alignment (Stage 1) + KL divergence (Stage 2) |

## C.3 MODEL ARCHITECTURES

Table 10 summarizes the model architectures. Asterisk (*) marks manually adjusted variants to simulate heterogeneous settings, covering diverse scales across both domains.

Table 10: Detailed architectural configurations of teacher and student models used in our experiments.

| Model | $d_{\text{model}}$ | $d_{\text{state}}$ | $d_{\text{conv}}$ | #Layers | #Params (M) | FLOPs (G) |
|---|---|---|---|---|---|---|
| **Vision Models** | | | | | | |
| Vim-Mini* | 96 | 16 | 4 | 12 | 1.1 | 0.15 |
| Vim-Mini | 96 | 16 | 4 | 24 | 2.1 | 0.28 |
| Vim-Tiny* | 192 | 16 | 4 | 12 | 3.8 | 0.55 |
| Vim-Tiny | 192 | 16 | 4 | 24 | 7.1 | 1.08 |
| Vim-Small | 384 | 16 | 4 | 24 | 26 | 4.00 |
| DeiT-Tiny | 192 | – | – | 12 | 5 | 1.26 |
| DeiT-Small | 384 | – | – | 12 | 22 | 4.61 |
| **Language Models** | | | | | | |
| DistilGPT2 | 768 | – | – | 6 | 88 | 21.68 |
| Phi-Mamba* | 768 | 64 | 4 | 6 | 123 | 20.76 |
| Phi1.5 | 2048 | – | – | 24 | 1418 | 336.11 |
| Phi-Mamba | 2048 | 64 | 4 | 24 | 1521 | 362.30 |

## C.4 TRAINING RECIPE

Tables 11 and 12 summarize our training recipes. Vision uses AdamW with cosine decay on 1–20% ImageNet, while language adopts a two-stage setup: 200M tokens for attention alignment and 2B/4B tokens for CE+KL distillation.

Table 11: Training recipe for visual distillation experiments under different data regimes.

| Config | 1% Data | 5% Data | 10% Data | 20% Data |
|---|---|---|---|---|
| Optimizer | AdamW | AdamW | AdamW | AdamW |
| Learning rate | 5e-4 | 5e-4 | 5e-4 | 5e-4 |
| Weight decay | 0.05 | 0.05 | 0.05 | 0.05 |
| Training epochs | 3000 | 600 | 300 | 150 |
| Optimizer momentum | (0.9, 0.999) | (0.9, 0.999) | (0.9, 0.999) | (0.9, 0.999) |
| Batch size | 64 | 64 | 64 | 64 |
| LR schedule | Cosine decay | Cosine decay | Cosine decay | Cosine decay |

# D DATASETS

In this section, we introduce the datasets used in the paper, including those for visual and language modeling.

**ImageNet-1K (Deng et al., 2009)** is a widely-used large-scale image classification dataset consisting of approximately 1.2 million training images and 50,000 validation images across 1,000 object categories. It serves as a standard benchmark for evaluating the performance of visual recognition models.

Table 12: Training recipe for language distillation experiments under different token budgets. Stage 1 uses only attention alignment; Stage 2 combines CE and KL distillation.

| Config | 200M Tokens (Stage 1) | 2B Tokens (Stage 2) | 4B Tokens (Stage 2) |
|---|---|---|---|
| Optimizer | AdamW | AdamW | AdamW |
| Learning rate | 2e-5 | 2e-5 | 2e-5 |
| Warmup ratio | 0.1 | 0.1 | 0.1 |
| Training objective | Attention alignment | CE + KL loss | CE + KL loss |
| per device Batch size | 32 | 32 | 32 |
| Max seq length | 1024 | 1024 | 1024 |

**OpenWebText (Gokaslan et al., 2019)** is a high-quality web text corpus constructed from URLs shared on Reddit posts with high karma. It captures diverse and natural language from the open web, making it well-suited for pretraining large language models.

**C4 (Colossal Clean Crawled Corpus) (Raffel et al., 2020)** is a large-scale English-language dataset created by filtering and cleaning the Common Crawl web archive. It is designed to provide high-quality, diverse text data for robust pretraining of general-purpose language models.

**WikiText (Merity et al., 2016)** WikiText is a collection of carefully curated, long-form Wikipedia articles that preserve document-level coherence. It is commonly used for benchmarking language models on tasks requiring contextual understanding and fluency.

## E  HEATMAP VISUALIZATION

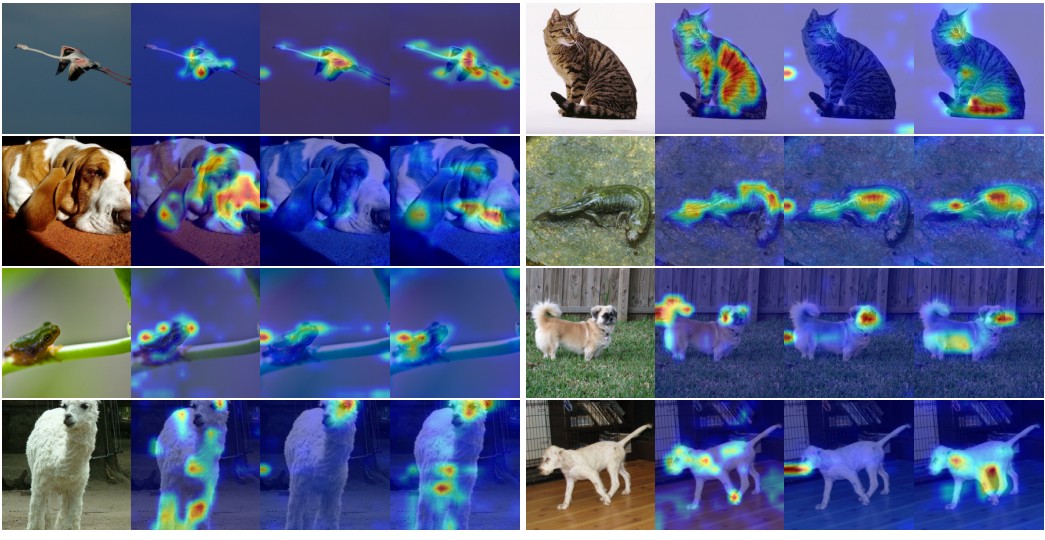

Figure 7: **Visualization of attention heatmaps across teacher and student models.** In each group of images, we show the original image followed by heatmaps generated from: (1) the pretrained ViT teacher, (2) a Vim model trained on the full ImageNet training set, and (3) our Vim student trained with CAB using only 10% of the full ImageNet training set. Despite limited supervision, our method produces more concentrated and semantically meaningful attention responses.

## F    LOSS TRAJECTORIES DURING DISTILLATION

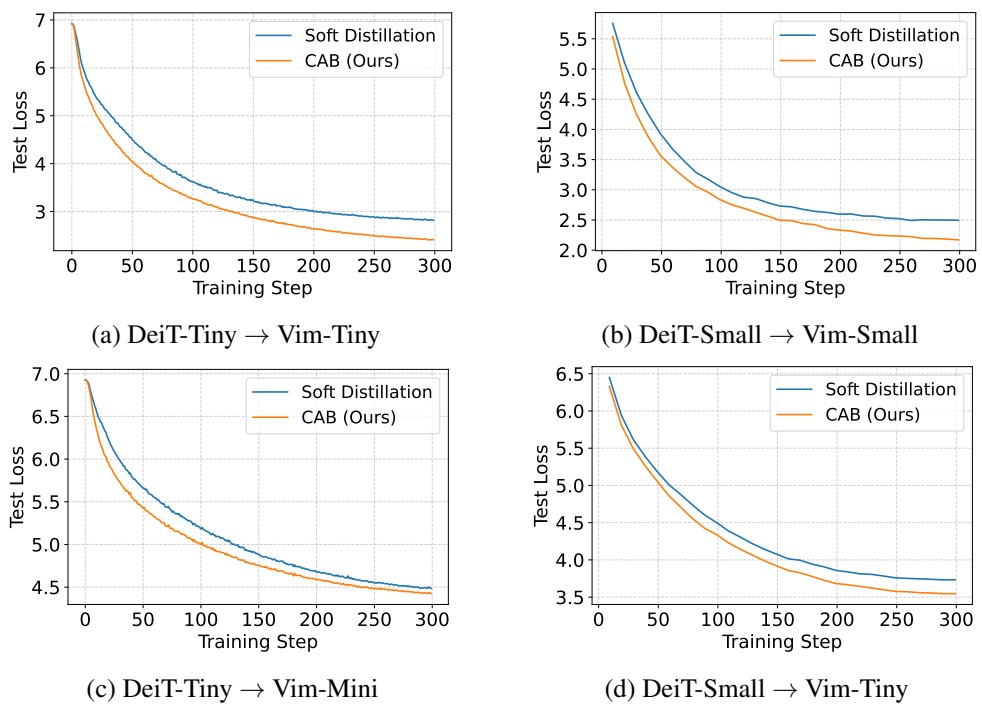

(a) DeiT-Tiny → Vim-Tiny

(b) DeiT-Small → Vim-Small

(c) DeiT-Tiny → Vim-Mini

(d) DeiT-Small → Vim-Tiny

Figure 8: Test loss trajectories of different teacher-student pairs across training. Each subfigure illustrates the effectiveness of our attention-level distillation under a specific architecture configuration.

## G    ACC TRAJECTORIES DURING DISTILLATION

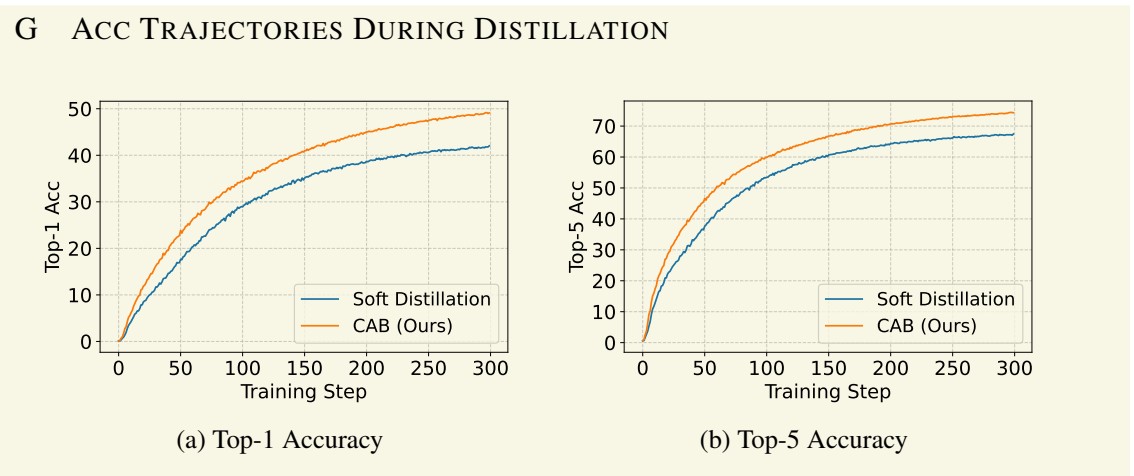

(a) Top-1 Accuracy

(b) Top-5 Accuracy

Figure 9: Test accuracy trajectories during the distillation process for **DeiT-Tiny → Vim-Tiny** using **10% ImageNet training data**. We report both Top-1 and Top-5 accuracy across training steps, showing that attention-level distillation consistently improves generalization throughout the entire training trajectory.

