# OpenReview forum: "Data Efficient Any Transformer-to-Mamba Distillation via Attention Bridge"
_ICLR.cc/2026/Conference — ICLR 2026 Conference Withdrawn Submission_

### Official Review · Reviewer_PJ6L · 2025-10-27

**Soundness:** 1
**Presentation:** 2
**Contribution:** 1
**Rating:** 2
**Confidence:** 4

**Summary:**

This paper presents a distillation method that transfers knowledge from Transformer based architectures to Mamba based models. Utilizing state space duality that's presented in Mamba2, the method is extremely simple -- attach additional MLP layers to B and C to predict K and Q. The authors validate their method mainly using vision models, ViT and Vim, and also showing some results in NLP.

**Strengths:**

The idea is very straightforward, which will be easy for readers to understand and implement. The authors provide many experiments using both vision and language data.

**Weaknesses:**

- Lack of novelty and contribution: Indeed, I am a fan of simple ideas. However, the results are quite suspicious due the following concerns;
- Unfair experimental setup: the paper uses \phi_B and \phi_C that are 2 layer MLP with SiLU. This leads to more parameters as well as more FLOPs. This is a huge benefit for the authors' method.
- Why vision? : ViT and Vim are fundamentally different that Vim rasterizes inputs, while there is no causal assumption for ViT. Also, it's hard to agree with the idea of matching both forward & backward B and C to K and V. It sounds weird as it will lead to forward and backward to be the same for B and C.

**Questions:**

Line 49 : What is "direct injecting Transformer attention into Mamba"?

---

> ### Author Response · Authors · 2025-11-19
>
> We sincerely thank the reviewer PJ6L for the feedback and suggestions.
> We provide our responses below and look forward to your further comments.
>
> > **W1:** Experimental setup.
>
> The projection MLPs introduce zero additional parameters to the student model. They are auxiliary training-time components that are fully removed after distillation. The final Vim model has exactly the same parameters, architecture, and inference FLOPs as the baseline.
>
> Using temporary projection modules is a common practice in cross-architecture alignment or distillation[1,2,3]. These methods add intermediate mappings during training but do not alter the final student model. CAB follows the same accepted paradigm.
>
> Empirically, CAB is in fact more efficient than existing SSM distillation frameworks: as shown in Fig.5, it achieves 10× lower memory usage and 4× faster runtime than the MOHAWK baseline. This shows that the projection introduces no extra burden and can even make training more efficient.
>
> > **W2:** Why vision?
>
> We acknowledge that Vim rasterizes inputs while ViT operates on a non-causal set of patches. However, this difference concerns the computational formulation, not the representational capacity.
>
> From a theoretical perspective, recent analyses [4] show that Mamba admits an attention-like linear formulation under its dual view. Although a single rasterized scan is causal, bidirectional scanning restores access to the full image context. Consequently, this reproduces the global non-causal receptive field of ViT with linear complexity.
>
> From an empirical perspective, works[5,6] demonstrate that the spatial inductive biases learned by ViTs can be successfully transferred into rasterized Mamba architectures. These results indicate that rasterization does not constrain what the model is able to represent, it only alters the mechanism through which interactions are computed.
>
> Therefore, the ViT→Vim setting is appropriate: rasterization is not a barrier to transferring non-causal visual knowledge.
>
> Importantly, we acknowledge that the description was unclear in the original version. We do not use the same bridge for the forward and backward directions. The B→Q and C→K mappings are implemented with two independent projection functions, each with separate parameters. This separation keeps Mamba’s forward and backward dynamics distinct instead of collapsing them into the same representation. We will clarify this more explicitly in the revised paper.
>
> > Q1: What is "direct injecting Transformer attention into Mamba"?
>
> “Direct injecting” refers to directly feeding the Transformer’s attention (from **Phi-1.5**) into the **Phi-Mamba** model[7]. Since Phi-Mamba is built on Mamba-2, whose State Space Duality rewrites the SSM update in a linear-attention form, we can substitute its internal interaction matrix with the teacher’s $QK^T$ attention map.
>
> **References:**
>
> [1] Romero, et al. FitNets: Hints for Thin Deep Nets, 2014.
>
> [2] Yu S, et al. Representation alignment for generation: Training diffusion transformers is easier than you think, 2024.
>
> [3] Chen P, et al. Distilling knowledge via knowledge review, 2021.
>
> [4] Ali A, et al. The hidden attention of mamba models, 2024.
>
> [5] Chen X, et al. Transmamba: Fast universal architecture adaption from transformers to mamba, 2025.
>
> [6] Wei G, Chellappa R. Vit-linearizer: Distilling quadratic knowledge into linear-time vision models, 2025.
>
> [7] Bick A, et al. Transformers to ssms: Distilling quadratic knowledge to subquadratic models, 2024.

---

> ### Author Response · Authors · 2025-11-27
>
> Thank you once again for your valuable feedback. We have added further explanations and clarifications to the paper based on your comments. Since the discussion phase has already begun, we would greatly appreciate knowing whether our responses have addressed your concerns. Your insights are extremely valuable to us, and we are very willing to address any remaining issues to further improve the work.

---

### Official Review · Reviewer_bjGZ · 2025-10-28

**Soundness:** 3
**Presentation:** 4
**Contribution:** 3
**Rating:** 6
**Confidence:** 4

**Summary:**

This paper introduces CAB (Cross-architecture distillation via Attention Bridge), a framework for distilling knowledge from Transformer-based teacher models to state-space model (SSM) students, specifically Mamba. The core innovation is a lightweight, MLP-based "Attention Bridge" that aligns the implicit attention carriers in Mamba (the token-dependent projections $B$ and $C$) with the explicit key ($K$) and query ($Q$) representations from the Transformer.

After the distillation process is complete, the MLP weights are discarded. The final product is a standalone Mamba model that has internalized the attention-based inductive biases from the Transformer teacher, with no inference-time overhead from the bridge components. The distillation process does not freeze the Mamba weights; instead, the gradients from the attention alignment loss and the standard output distillation loss are backpropagated through the MLP bridge and into all weights of the student model.



This method enables fine-grained, token-level supervision without the quadratic memory cost of aligning full attention matrices. It also employs a flexible, proportional layer-wise alignment strategy to handle architectural depth mismatches. The authors demonstrate CAB's effectiveness and data efficiency through extensive experiments on both vision (ImageNet) and language (OpenWebText, C4, WikiText) tasks, showing consistent improvements over standard and recent cross-architecture distillation baselines, particularly in low-data regimes.

**Strengths:**

* The formulation of the "Attention Bridge" is a creative and novel solution to the cross-architecture distillation problem. The insight to treat Mamba's $B$ and $C$ projections as analogous to the Transformer's $K$ and $Q$ under a linear attention approximation is theoretically grounded and is a natural extension of State-Space Duality.

* The experimental evaluation is thorough, spanning multiple domains, model scales, and data regimes. The consistent and often substantial improvements over strong baselines is very convincing. The ablation studies are well-designed and provide crucial insights into the contribution of each component.

* The paper is well-written. The problem is motivated, the method is explained with sufficient detail, and the results are presented logically and convincingly.

* This work has high practical significance. By enabling data-efficient knowledge transfer from Transformers to Mamba, it lowers the barrier to adopting high-performance SSMs in real-world scenarios where data is limited and computational resources are constrained.

**Weaknesses:**

* The authors use a 2-layer MLP with SiLU activations for the bridge modules ($\phi_B$) and ($\phi_C$). While effective, the paper does not ablate this choice against a simpler linear projection. A brief justification or experimental result showing the superiority of a non-linear transformation over a linear one would strengthen this design decision, especially given the focus on efficiency.

* The proportional layer mapping $g(l)$ is a simple and effective heuristic. However, the paper does not explore or ablate against other possible strategies (e.g., learned alignment, manual assignment based on layer depth). A brief discussion on the choice and potential alternatives would be useful.

* A key, unexplored trade-off is whether distilling a Transformer's inductive biases into a Mamba model also transfers the Transformer's weakness in length generalization. State-space models like Mamba are prized for their ability to generalize to sequences longer than those seen in training, a known challenge for Transformers. The paper lacks any analysis or discussion of whether the distilled models retain their robustness to longer sequence lengths.

**Questions:**

* Why was a 2-layer MLP chosen for the bridge instead of a simpler linear projection? Was this choice ablated?

* Does the CAB method require the teacher and student to share the same tokenizer?

---

> ### Author Response · Authors · 2025-11-19
>
> We sincerely thank the reviewer bjGZ for recognizing the novelty of our Attention Bridge formulation and for the constructive suggestions. We provide our responses below.
>
>
> > **W1 & Q1:** Lack of systematic validation for the bridge design.
>
> To address the concern, we performed additional controlled ablations on the architecture of the attention bridge, varying the network depth.
>
> **Exp: Bridge Type Ablation**
>
> **Setting:**
>
> All models are trained for 300 epochs on a 10% ImageNet-1K subset, using lr = 5e-4 and batch size = 64.
> The teacher and student models are DeiT-Tiny → Vim-Tiny.
>
> We ablate the bridge architecture by comparing **a linear projection, a 1-layer MLP, a 2-layer MLP, and a 3-layer MLP**, and all MLP-based variants use ReLU as the activation function.
>
> **Results:**
>
> | Bridge Variant        | acc@Top-1 |
> |-----------------------|-----------|
> | Linear projection     |   45.0    |
> | 1-layer MLP           |   48.6    |
> | 2-layer MLP (default) |   49.2    |
> | 3-layer MLP           |   49.3    |
>
> **Conclusion:**
> The results show that a purely linear projection performs significantly worse, indicating that non-linear mappings are essential. While deeper MLPs (3-layer) offer slight improvements over the 2-layer variant, the gain is marginal relative to the additional parameters and computation. Thus, the 2-layer MLP offers the best efficiency–performance trade-off and is used as the default bridge design.
>
> > **W2：Ablation about cross-layer alignment strategies**
>
> To further examine mismatched-depth cases (e.g., teacher = 12 layers, student = 24 layers), we evaluated two alternative strategies:
> (1) mapping all teacher layers to the **first 12 student layers**, and
> (2) mapping all teacher layers to the **last 12 student layers**.
>
> **Exp: Cross-Layer Alignment Ablation**
>
> **Setting:**
> All models are trained for 300 epochs on 10% ImageNet-1K, using lr = 5e-4 and batch size = 64.
> Teacher–student pair: DeiT-Small(12 layer) → Vim-Tiny(24 layer).
>
> **Results:**
>
> | Alignment Strategy                       | acc@Top-1 |
> |------------------------------------------|-----------|
> | Teacher → Student (first 12 layers)      |   44.1    |
> | Teacher → Student (last 12 layers)       |   43.6    |
> | Proportional mapping (ours)              |   44.9    |
>
> **Conclusion:**
> Both fixed “first-12” and “last-12” mappings yield lower performance than proportional mapping.
> Consistent with prior work[1,2], the proportional scheme remains the most stable and robust alignment method, especially when teacher and student depths differ.
>
> > **W3：** Distilling Transformer inductive biases might weaken Mamba’s natural ability to generalize to longer sequences.
>
> We thank the reviewer for the insightful question.
>
> Mamba’s ability to generalize to long sequences mainly comes from its state matrix \(A\) and its continuous-time recurrent update rule. These components control how information is propagated over long contexts. CAB only aligns the token-level projection space, that is, \(B/C\) with \(Q/K\). It does not modify the state matrix \(A\) or the recurrent dynamics that support long-sequence extrapolation.
> Therefore, CAB provides useful early structural guidance while keeping the part of the model responsible for length generalization unchanged.
>
> We will clarify this point in the revised version.
>
>
>
>
> > **Q2:** Tokenizer Setting.
>
> In our LM experiments, the teacher and student use the same tokenizer.
>
>
> **References:**
>
> [1] Jiao X, et al. Tinybert: Distilling bert for natural language understanding, 2020.
>
> [2] Yu Z, et al. Revisiting Intermediate-Layer Matching in Knowledge Distillation: Layer-Selection Strategy Doesn't Matter (Much), 2025.

---

> ### Author Response · Authors · 2025-11-27
>
> Thank you once again for your valuable feedback. We have added further explanations and clarifications to the paper based on your comments. Since the discussion phase has already begun, we would greatly appreciate knowing whether our responses have addressed your concerns. Your insights are extremely valuable to us, and we are very willing to address any remaining issues to further improve the work.

---

> > ### Comment · Reviewer_bjGZ · 2025-11-27
> >
> > I thank the authors for the additional experiments and answering my questions. I had no major concerns in my initial review and I will keep my score unchanged.

---

### Official Review · Reviewer_BZVQ · 2025-10-31

**Soundness:** 3
**Presentation:** 3
**Contribution:** 3
**Rating:** 6
**Confidence:** 4

**Summary:**

This paper addresses the challenge of cross-architecture knowledge distillation from pretrained Transformers to state-space models (SSMs), a critical need as SSMs offer superior runtime efficiency for sequence modeling but suffer from high training costs, immature ecosystems, and structural heterogeneity with Transformers. To solve this, the authors propose Cross-architecture distillation via Attention Bridge (CAB), a data-efficient framework that enables effective transfer of attention-based knowledge from Transformers to SSMs, even under limited training data. CAB overcomes these challenges through these key components: 1. Lightweight MLP attention bridge, Maps Transformer’s explicit attention carriers (\(Q, K\)) to Mamba’s implicit attention projections (\(B, C\)) via learnable MLPs (\(\phi_B, \phi_C\)). 2. Flexible layer-wise alignment. 3. Training framework: First aligns attention via the bridge (200M tokens), then performs soft KD (KL divergence) for output refinement (2B/4B tokens).

**Strengths:**

1. Innovative cross-architecture distillation paradigm with structural alignment.

The paper demonstrates strong originality through creative solutions to long-standing cross-architecture knowledge transfer challenges. CAB introduces the Attention Bridge—a lightweight MLP-based module that maps Transformer’s explicit Q/K representations to Mamba’s implicit B/C projections—addressing the core issue of structural heterogeneity between attention-based models and SSMs. Unlike prior works, this design avoids quadratic overhead while enabling fine-grained token-level supervision.

2. Comprehensive, and convincing experimental validation.

The experimental quality is exemplary, characterized by thoroughness, reproducibility, and statistically significant results. Across two distinct domains (vision with ImageNet-1k, language with OpenWebText/C4/WikiText), the method consistently outperforms baselines.

3. Well-structured narrative with transparent method and experimental details.

The paper excels in clarity, with a logical flow that guides readers from problem to solution. The problem formulation is concise: it first highlights SSMs’ efficiency but high training costs, then critiques limitations of existing distillation (weak gradient signals, architectural mismatch), and motivates CAB’s design through visual evidence (Fig. 1 showing performance collapse from direct attention injection).

4, Addressing critical gaps and advancing efficient sequence modeling.

The paper’s significance is far-reaching, both for research and practical applications. From a research perspective, it bridges the divide between Transformer’s rich pretrained knowledge and SSMs’ efficient inference, enabling SSMs to leverage the mature Transformer ecosystem—this opens new avenues for cross-architecture distillation beyond Transformer-to-Transformer or CNN-to-Transformer paradigms. Practically, the method’s data efficiency (effective with 1–20% training data) addresses real-world constraints in low-resource domains (healthcare, robotics, edge computing), where large datasets are scarce. Furthermore, it provides a generalizable framework for transferring attention-based inductive biases to recurrent models, inspiring future research on distillation between heterogeneous sequence modeling architectures.

**Weaknesses:**

1. Empirical Considerations for Attention Bridge Design: Lack of Systematic Validation

The paper designs the attention bridge as a "2-layer MLP + SiLU activation," but this design lacks theoretical basis and systematic comparison: (1) It does not test the impact of different network structures (1-layer MLP, 3-layer MLP, attention layer, etc.) on the alignment effect; (2) It does not analyze the trade-off between the bridge's parameter size and performance efficiency, if the bridge's parameters are reduced (e.g., 1-layer MLP), can the alignment effect still be maintained? If the parameters are increased (e.g., 3-layer MLP + attention mechanism), can performance be further improved without significantly increasing computational cost?

2. Performance Degradation in Full Data Scenarios: Unresolved Core Contradictions

Appendix A.1 of the paper shows that when training on full data for 300 epochs, the Top-1 accuracy of CAB (71.34%) is lower than that of soft distillation (74.56%). This is only alleviated by “early stopping alignment (switching to KL distillation after 35/50 epochs)”, but the reasons for the degradation and the fundamental solutions are not analyzed in depth: (1) It is not clear whether it is “over-constraining of the student model by attention supervision” or “the bridge structure introducing redundant information in full data”; (2) The early stopping strategy lacks an adaptive mechanism – how to determine the optimal early stopping epoch under different tasks and different teacher-student model combinations? Manually setting 35/50 epochs lacks universality.

3. The Performance of Scaled to Large Models is unknown.

As shown in the experimental section of the paper, most of the models used in the experiments are limited to below M. Is CAB distillation efficient and performant enough, for example, using 1B, 7B as the teacher, or a larger model as the teacher or student? Is it possible that the mapping effect would be affected if a large model is used as the student?

**Questions:**

1. Choice of MLP Projection in Attention Bridge: Why use a 2-layer MLP with SiLU activation for φ_B and φ_C? Did you ablate other architectures (e.g., linear projections, 1-layer MLPs, different activations) and find the 2-layer MLP to be uniquely effective?

2. Layer Mapping Function g(l): The paper uses proportional indexing for layer alignment (g(l) = ⌊l/L · T⌋). Did you test alternative alignment strategies (e.g., learned layer mapping, fixed one-to-one for matching depths) and find proportional indexing to be superior?

3. Early-Stopped Alignment in Full-Data Regime: Appendix A.1 shows that CAB degrades performance if applied for 300 epochs in full-data settings, requiring early stopping. Why does excessive attention supervision harm performance? Is this related to overfitting to Transformer inductive biases that are suboptimal for Mamba in full-data scenarios?


4. Data Scarcity Boundaries: The paper tests 1–20% of ImageNet and 200M–4B tokens for language. What is the minimum data threshold where CAB outperforms baselines? For example, does CAB still help with 0.5% of ImageNet, or does it fail due to insufficient supervision for the attention bridge? Or maybe would it affects the performance if test 40% or more part of data?


5. Why Not Use Learned Activation Mappings for Linear Attention: Linear attention uses φ(Q) and φ(K) for efficiency. Did you consider aligning Mamba’s B/C to φ(K)/φ(Q) (instead of raw K/Q) to leverage linear attention’s structural similarity to Mamba?

---

> ### Author Response · Authors · 2025-11-19
>
> We sincerely thank Reviewer BZVQ for acknowledging the strengths of our method and for the constructive suggestions. We make responses as follows.
>
> > **W1 & Q1:** Lack of systematic validation for the bridge design.
>
> **A1:** To address the concern, we performed additional controlled ablations on the architecture of the attention bridge, varying both network depth and activation functions.
>
> **Exp1: Bridge Type Ablation**
>
> **Setting:**
>
> All models are trained for 300 epochs on a 10% ImageNet-1K subset, using lr = 5e-4 and batch size = 64.
> The teacher and student models are DeiT-Tiny → Vim-Tiny.
>
> We ablate the bridge architecture by comparing **a linear projection, a 1-layer MLP, a 2-layer MLP, and a 3-layer MLP**, and all MLP-based variants use ReLU as the activation function.
>
> **Results:**
>
> | Bridge Variant        | acc@Top-1 |
> |-----------------------|-----------|
> | Linear projection     |   45.0    |
> | 1-layer MLP           |   48.6    |
> | 2-layer MLP (default) |   49.2    |
> | 3-layer MLP           |   49.3    |
>
> **Conclusion:**
> The results show that a purely linear projection performs significantly worse, indicating that non-linear mappings are essential. While deeper MLPs (3-layer) offer slight improvements over the 2-layer variant, the gain is marginal relative to the additional parameters and computation. Thus, the 2-layer MLP offers the best efficiency–performance trade-off and is used as the default bridge design.
>
> **Exp2: Activation Function Ablation**
>
> **Setting:**
>
> Same training configuration as Exp1.
> We fix the bridge to a **2-layer MLP** and vary activation functions: **ReLU**, **GELU**, **SiLU**.
>
> **Results:**
>
> | Activation     | acc@Top-1 |
> |----------------|-----------|
> | ReLU           |    49.1   |
> | GELU           |    49.2   |
> | SiLU (default) |    49.2   |
>
> **Conclusion:**
> All three activation functions produce nearly identical performance, showing that CAB is not sensitive to the specific choice of activation. We retain **SiLU** as the default, following prior works [1] that adopt SiLU for stable optimization in lightweight projection modules.
>
>
> > **W2 & Q3:** Performance degradation in the full-data regime.
>
> **A2:** We thank the reviewer for raising this important point.
> Our analysis indicates that the degradation originates from over-constraining the student rather than from any redundancy introduced by the bridge.
> Prior work on representation alignment[2] shows that strong alignment signals are helpful early in training but become suboptimal if enforced for the entire training trajectory.
>
> This matches our observations: CAB significantly improves early-stage learning, but prolonged attention supervision limits the student SSM’s ability to develop its own inductive biases in the late stage. The effectiveness of early-stopped alignment supports this explanation: applying CAB only in the early phase provides useful structural guidance while preserving the student’s flexibility to learn its own representations later. This shows that the bridge is not harmful, only overly prolonged alignment is.
> Exploring an adaptive stopping rule is an interesting direction, we leave this for future work as it requires task-specific dynamic criteria beyond the scope of the current paper.
>
> > **W3:** Concern about the scalability of CAB to larger model sizes.
>
> **A3:** In the supplementary A.2 LM experiments, we already scale the framework to the 1B level, demonstrating that CAB remains stable and effective at substantially larger model sizes.
> This large-scale configuration is strictly aligned with prior studies[3] that evaluate representation alignment at the billion-parameter scale, ensuring a fair and consistent comparison.

---

> > ### Author Response · Authors · 2025-11-19
> >
> > > **Q2** Ablation about cross-layer alignment strategies.
> >
> > **A4:** We emphasize that our proportional layer mapping naturally reduces to **one-to-one alignment when the teacher and student share the same depth**.
> > To further examine mismatched-depth cases (e.g., teacher = 12 layers, student = 24 layers), we evaluated two alternative strategies:
> > (1) mapping all teacher layers to the **first 12 student layers**, and
> > (2) mapping all teacher layers to the **last 12 student layers**.
> >
> > **Experiment: Cross-Layer Alignment Ablation**
> >
> > **Setting:**
> > All models are trained for 300 epochs on 10% ImageNet-1K, using lr = 5e-4 and batch size = 64.
> > Teacher–student pair: DeiT-Small(12 layer) → Vim-Tiny(24 layer).
> >
> >
> > **Results:**
> >
> > | Alignment Strategy                       | acc@Top-1 |
> > |------------------------------------------|-----------|
> > | Teacher → Student (first 12 layers)      |   44.1    |
> > | Teacher → Student (last 12 layers)       |   43.6    |
> > | Proportional mapping (ours)              |   44.9    |
> >
> > **Conclusion:**
> > Both fixed “first-12” and “last-12” mappings yield lower performance than proportional mapping.
> > We did not adopt learned mapping functions because they introduce additional parameters and undermine CAB’s **lightweight** design philosophy.
> > Consistent with prior work[4,5], the proportional scheme remains the most stable and robust alignment method, especially when teacher and student depths differ.
> >
> >
> > > **Q4: Data-scarcity boundaries and behavior under extremely low or high data regimes**
> >
> > **A5:** To investigate how CAB behaves outside the 1–20% data range, we conducted an additional experiment at 0.5% ImageNet, an extremely scarce-data setting.
> >
> > **Data Fraction Ablation (0.5% ImageNet)**
> > All models are trained for **300 epochs** on **0.5% ImageNet-1K**, using **lr = 5e-4** and **batch size = 64**.
> > Teacher–student pair: **DeiT-Tiny → Vim-Tiny**.
> > We compare Vanilla training, Soft Distillation, and CAB.
> >
> > **Results:**
> >
> > | Method           | acc@Top-1 |
> > |------------------|-----------|
> > | Vanilla          |   2.71    |
> > | Soft Distill     |   4.26    |
> > | **CAB (ours)**   |  **5.64** |
> >
> >
> > **Conclusion:**
> > CAB remains highly effective even at 0.5% data, as the alignment provides useful structural supervision that compensates for extreme data scarcity. In contrast, in full-data settings prolonged alignment may over-constrain the student (see Reply for W2 & Q3), suggesting that CAB is most beneficial in low-data regimes and should be early-stopped when data are abundant. We will add this analysis in the revised version.
> >
> >
> >
> > > **Q5:** Why not align Mamba’s B/C to Φ(K)/Φ(Q)?
> >
> > **A6:** Our method initially started from this direction — aligning **B/C → Φ(K)/Φ(Q)**.  However, the attention space (Q/K or φ(Q)/φ(K)) has a **much larger hidden dimension** than the Mamba state size d_state. In practice, compressing Q/K into the much smaller B/C space consistently led to worse performance, as the student loses too much information during compression. By contrast, projecting B/C into the higher-dimensional Q/K space provides the student with **richer representational capacity** and achieved **better empirical results**. Therefore, we adopt **Φ(B)/Φ(C) → K/Q** alignment.
> >
> >
> > **References:**
> >
> > [1] Yu S, et al. Representation alignment for generation: Training diffusion transformers is easier than you think, 2024.
> >
> > [2] Wang Z, et al. REPA Works Until It Doesn't: Early-Stopped, Holistic Alignment Supercharges Diffusion Training, 2025.
> >
> > [3] Bick A, et al. Transformers to ssms: Distilling quadratic knowledge to subquadratic models. 2024.
> >
> > [4] Jiao X, et al. Tinybert: Distilling bert for natural language understanding, 2020.
> >
> > [5] Yu Z, et al. Revisiting Intermediate-Layer Matching in Knowledge Distillation: Layer-Selection Strategy Doesn't Matter (Much), 2025.

---

> ### Author Response · Authors · 2025-11-27
>
> Thank you once again for your valuable feedback. We have added further explanations and clarifications to the paper based on your comments. Since the discussion phase has already begun, we would greatly appreciate knowing whether our responses have addressed your concerns. Your insights are extremely valuable to us, and we are very willing to address any remaining issues to further improve the work.

---

### Official Review · Reviewer_r1ob · 2025-11-01

**Soundness:** 2
**Presentation:** 2
**Contribution:** 2
**Rating:** 2
**Confidence:** 4

**Summary:**

The paper presents a method to distill a pretrained transformer model into a student Mamba model. The core idea of the paper relies on the high-level shared functionality of the B and C projections of the SSM and the K and Q projections of the Transformer. The distillation method aligns these using a small learnable MLP projection module (the Attention Bridge). The alignment can also happen in situations where the number of layers in the student and teacher models is different by doing proportional indexing between layers in the learning process.

Overall, I found that the paper has a combination of neat ideas, however is seriously lacking in experimental rigor as well as soundness (see weaknesses and questions). It also makes some claims that I believe are misleading. In the current state, I am leaning to reject the paper.

**Strengths:**

- The paper has a combination of intuitive ideas (e.g. the Attention Bridge, the layerwise proportional indexing for asymmetric teacher and students), and these are introduced and demonstrated well.

- The chosen baselines make sense to compare against, and the proposed method seems generally superior to them.

- The paper is generally well written and is easy to read and understand.

**Weaknesses:**

- The paper restricts evaluation to situations in which the choice of teacher model is the same (or similar) size as the student model (e.g. Deit Tiny to Vim Tiny, Deit Small to Vim Small in most experiments, and only one experiment with the immediately larger model variation used as teacher). I think that distillation presents the most practical value when we can use a larger teacher to train a smaller student model (e.g. Deit base, large, or huge used as a teacher for a smaller Vim model). The paper contains a layer-wise alignment strategy in L216 which was used for the other comparisons, but not for the situation in which the teacher model is significantly larger (which is often going to be the case in a practical distillation scenario)

- I find it problematic that we need to do 300 epochs of distillation on a randomly chosen subset of ImageNet, and the only reported metric is top-1 ImageNet accuracy. Moreover, we use a teacher model trained on ImageNet to train a student model that is also previously trained on ImageNet!

- ImageNet-1K as a dataset in itself has several unique characteristics (e.g. artificially balanced classes, a particular kind of class distribution) etc that are usually not representative of real deployments. In my opinion, showing some results on additional tasks (e.g. ADE-20k segmentation like the ViM paper does) would have been helpful to strengthen the paper.

- I find Figure 1 (a) slightly misleading for a number of reasons. For example, calling SSM-based training slow and attention based inference slow are both a massive stretch. The real inference performance of Transformers with FlashAttention etc can be faster than supposedly linear SSM implementations. At the same time, the training time of a Mamba-2 model can be lesser than an equivalent transformer due to faster convergence. Making such a strong claim in the most important figure of the paper could mislead the reader.

- I think the LM side of the experiments could have used significantly more rigor. E.g. using additional base models at different parameter ranges, and evaluating on additional downstream tasks using a standard tool like LM Eval Harness. In general, the core result is just comparing DistillGPT to PhiMamba, which is very insufficient in my opinion.

- The choice of LM models was conceivably due to a similar number of GFlops. However, this alone can be misleading given the different execution methods of the Transformer and Mamba models. I also find it slightly problematic that an 88M param model was used as a teacher for a 123M param model. At this model size, there is nothing concrete that can be ascertained about scaling trends, but at the very least, I believe that the teacher model here could have been a few more variations with higher parameter counts.

- The student models in the paper are typically also pretrained models on a specific dataset. For a more generally applicable method, I think there should also have been results for when the student model is trained from scratch and does not contain the same data biases that the teacher model does.

**Questions:**

- For the claim in the abstract, “Moreover, the structural heterogeneity between
SSMs and Transformers makes it challenging to efficiently distill knowledge from
pretrained attention model” -> how do you reconcile this with a work like Mamba-2 showing the structured state space duality between Transformers and Mamba-2 like models?

- Why is Section A.2 in the appendix and not the main paper? Results on a 1.5B model are a lot more relevant than the 123M parameter model in my opinion. Also, why not additional LM parameter variations?

- “Standard ImageNet augmentations, including random cropping and horizontal flipping, are applied during training” -> this is often a critical detail in ImageNet runs. The chosen baseline method (DeiT) and the Vision Mamba model use slightly different ones, which may lead to inconsistent behavior. Could you please specify the exact augmentations and training parameters used?

---

> ### Author Response · Authors · 2025-11-19
>
> We sincerely thank Reviewer r1ob for the constructive comments and suggestions on extra experiments. We make responses as follows.
>
> > **W1:** Use of small-scale teacher limits empirical strength.
>
> **A1:**
> We appreciate the reviewer’s concern and provide clarifications on our framework below.
>
> First, existing Transformer to Mamba distillation approaches (e.g., [1][2]) typically require the teacher and student to have the same number of layers, which severely limits their applicability in realistic scenarios. Addressing this limitation is one of the main motivations of our work. Therefore, to provide fair and directly comparable baselines, our main experiments follow the same matched-depth setting used in prior approaches.
>
> At the same time, we fully agree that practical distillation often involves a significantly larger teacher guiding a smaller student. For this reason, we include the experiments in Table 6, where reduced-depth and reduced-width Vim students are distilled from substantially larger DeiT teachers. In these heterogeneous-scale settings, our method also shows clear improvements over traditional distillation baselines, and these results are enabled by our layer-wise alignment strategy (L216), which effectively handles the significant depth and width mismatch between the teacher and student.
>
>
> > **W2 & W7:** 300-epoch training on an ImageNet subset, the use of only Top-1 accuracy, and a concern about the distillation setup.
>
> **A2:** To avoid misunderstanding, we clarify that all student models in our distillation experiments are trained from random initialization. Accordingly, the 'vanilla training' baseline in Table 3 also trains the same randomly initialized student on the same dataset, but with only cross-entropy loss and no distillation signals. We will make this distinction clearer in the revised version.
>
> The choice of 300 epochs follows the default DeiT/Vim training protocol and ensures consistency with standard ImageNet practice when training from scratch.
>
> Regarding evaluation, Top-1 accuracy is the primary metric used across DeiT/Vim literature, but we agree broader metrics are helpful. In the revised paper (Appendix G), we now include Top-1 / Top-5 test accuracy trajectories throughout training to provide a more complete characterization of the distillation process.
>
> >**W3:** ImageNet-1K may not reflect real deployment distributions; additional tasks would strengthen the paper.
>
> **A3:** We agree that evaluating beyond ImageNet-1K can further strengthen the empirical validation.
>
> To provide a fast yet meaningful check within the rebuttal timeline, we first extract a 5% ImageNet-1K subset, and then apply **Pareto sampling (power = 6)** to the training split to create a realistic long-tailed distribution, following the protocol in [1]. The test set remains balanced. These results show that our method remains consistently effective even under highly imbalanced and more realistic scenarios, complementing the balanced ImageNet-1K setting.
>
> **Setting:**
>
> All models are trained for 300 epochs on the **5% long-tailed ImageNet subset** using a learning rate of 5e-4 and a batch size of 64.
> The teacher and student models are DeiT-Tiny → Vim-Tiny.
>
> **Results:**
>
> | Method          | acc@Top-1 | acc@Top-5 |
> |-----------------|-----------|-----------|
> | Vanilla Training |   3.69    |   9.13    |
> | SoftDistill     |   4.52    |   11.50   |
> | **CAB(ours)**   |  **5.56** |   **14.11**   |
>
>
> While implementing a full ADE20K segmentation pipeline would require substantial task-specific engineering and is unfortunately beyond the scope of the rebuttal timeline, we acknowledge its value and will highlight it as an important direction for future extension.
>
> > **W4:** Figure may be misleading.
>
>
> **A4:** We sincerely thank the reviewer for this insightful suggestion. We agree that the description in Figure 1(a) is overly simplified and may be misleading.
>
> Our original intent was to illustrate basic algorithmic differences.
> SSMs have O(n) recurrent computation, while Transformers have O(n²) attention computation.
> We also wanted to highlight that Transformers support highly efficient parallel training.
> In contrast, SSMs rely on parallel scans, which is an operation that is memory-bound and less efficient on current hardware.
>
> We have revised the figure and caption in the updated manuscript to more accurately reflect this distinction and avoid potential misunderstanding.

---

> > ### Author Response · Authors · 2025-11-19
> >
> > > **W5 & W6 & Q2:** Concerns about LM experimental rigor and the choice of model sizes.
> >
> > **A5:** As we noted in our response to W1, state-of-the-art LM distillation baselines require strict layer-to-layer structural alignment between teacher and student. To ensure controlled and fair comparisons, our LM experiments therefore adopt matched-depth teacher–student pairs, which necessarily constrains the range of possible model scales.
> >
> > Beyond this, we selected DistillGPT2 not only because its architecture matches the alignment requirements of prior baselines, but also because it is a well-established OpenWebText-trained setting that supports both in-domain and out-of-domain evaluation. This allows us to perform a clean and interpretable comparison under standardized LM training conditions.
> >
> > Regarding scale, we additionally include a 1.5B-parameter experiment in Appendix A.2. This result confirms that our method remains effective from ~100M up to 1.5B parameters, validating the scalability of the framework.
> > We placed these experiments in the appendix due to space constraints, while keeping the main text focused on the core theoretical contribution.
> >
> > > **Q1:** How does the claim on structural heterogeneity reconcile with Mamba-2’s Transformer–SSM duality.
> >
> > **A6:** The abstract refers to the architectural mismatch between pretrained Transformers and SSMs, especially the gap between Q/K/V projections and the (A, B, C) state operators.
> > This gap makes direct distillation difficult. The duality in Mamba-2 describes function-level equivalence, but it does not mean that Transformer parameters are structurally compatible with SSM parameters.
> > Expressive alignment does not eliminate the representation gap between attention heads and state-space dynamics.
> >
> > As noted in L292, our LM experiments already use Mamba-2, so our method is evaluated within the duality framework.
> > However, duality alone does not provide a practical mapping from Transformer weights to Mamba-2 weights.
> > An explicit alignment mechanism is still needed, and this is exactly what CAB provides.
> >
> > Thus, our claim about “structural heterogeneity” is not contradictory to Mamba-2; instead, the duality further motivates the need for practical alignment methods like ours.
> >
> > > **Q3:** concern about the precise augmentation pipeline and training hyperparameters.
> >
> > **A7:** To avoid ambiguity, we clarify that all ImageNet experiments strictly follow the official Vision Mamba training framework, which directly inherits the DeiT training pipeline and data augmentations.
> > This is also why we adopt DeiT as the teacher model.
> > Specifically, our experiments use RandomResizedCrop, RandomHorizontalFlip, Mixup (α = 0.8), and CutMix (α = 1.0),identical to the default DeiT/Vim augmentation pipeline.
> > The complete training configuration is provided in Appendix C.4 (Table 11).
> >
> > **References:**
> >
> > [1] Bick A, et al. Transformers to ssms: Distilling quadratic knowledge to subquadratic models, 2024.
> >
> > [2] Wang J, et al. The mamba in the llama: Distilling and accelerating hybrid models, 2024.
> >
> > [3] Liu Z, et al. Large-scale long-tailed recognition in an open world, 2019.

---

> ### Author Response · Authors · 2025-11-27
>
> Thank you once again for your valuable feedback. We have added further explanations and clarifications to the paper based on your comments. Since the discussion phase has already begun, we would greatly appreciate knowing whether our responses have addressed your concerns. Your insights are extremely valuable to us, and we are very willing to address any remaining issues to further improve the work.

---

### Note · Authors · 2026-01-26

I have read and agree with the venue's withdrawal policy on behalf of myself and my co-authors.

---

### Meta-Review · Area_Chair_Xwjx · 2026-01-08

**Summary:**

This paper proposes CAB, a distillation method from Transformers to SSMs via an attention bridge. While the idea is intuitive and the authors provide extensive responses, my recommendation is rejection. The decision stems from two primary concerns raised by reviewers. First, the novelty and significance of the contribution are questioned (PJ6L, r1ob), with critiques that the core idea is straightforward and its necessity in light of SSM-Transformer duality is unclear. Second, the experimental rigor and scope are insufficient. Key limitations include the lack of large-scale teacher-student evaluations, inadequate analysis of performance degradation in full-data regimes, and an incomplete exploration of the method's impact on SSMs' core strengths like length generalization. Although the rebuttal adds valuable ablations, it does not fully resolve these foundational issues regarding the paper's contribution and completeness.

**Reviewer Concerns:**

The rebuttal addressed several specific technical concerns: it provided ablations for the bridge design (BZVQ, bjGZ), clarified that projection modules are training-only (PJ6L), added experiments on data-scarce (0.5%) and long-tailed settings (r1ob), and discussed scalability to 1.5B models (BZVQ, r1ob). However, major concerns remain outstanding: 1) Limited novelty and a lack of compelling justification for the method's necessity beyond prior duality insights (PJ6L, r1ob's Q1). 2) Incomplete empirical validation, including the need for larger-scale teacher-student pairs, a principled solution—not just early stopping—for full-data performance drop, and an analysis of length generalization (r1ob, BZVQ, bjGZ). 3) Unresolved questions about the fundamental experimental setup and the validity of the vision-domain comparison remain (PJ6L).

**Reviewer Scores:**

If fully engaged, r1ob (2) might have raised the score to a 3, as the rebuttal added experiments on data regimes and scaling, but core issues about practical utility and rigorous LM evaluation likely persist. PJ6L (2) might also have moved to a 4, as concerns about experimental fairness were clarified, but doubts about novelty and architectural comparison likely remain. BZVQ (6) could have increased to a 7, as thorough ablations addressed specific design choices, though the full-data performance issue is not fully resolved. bjGZ (6) might have also moved to a 7, with bridge ablations provided and a reasonable explanation given for length generalization, leaving no major outstanding concerns.

---

### Decision · Program_Chairs · 2026-01-26

Reject